# Scaling Robot Policy Learning via Zero-Shot Labeling with Foundation Models

**Nils Blank**[1]    **Moritz Reuss**[1]    **Marcel Rühle**[1]    **Ömer Erdinç Yağmurlu**[1]
**Fabian Wenzel**[1]    **Oier Mees**[2]    **Rudolf Lioutikov**[1]

[1]Karlsruhe Institute of Technology    [2]UC Berkeley

**Abstract:** A central challenge towards developing robots that can relate human language to their perception and actions is the scarcity of natural language annotations in diverse robot datasets. Moreover, robot policies that follow natural language instructions are typically trained on either templated language or expensive human-labeled instructions, hindering their scalability. To this end, we introduce **NILS**: **N**atural language **I**nstruction **L**abeling for **S**calability. NILS automatically labels uncurated, long-horizon robot data at scale in a zero-shot manner without any human intervention. NILS combines pretrained vision-language foundation models in order to detect objects in a scene, detect object-centric changes, segment tasks from large datasets of unlabelled interaction data and ultimately label behavior datasets. Evaluations on BridgeV2, Fractal and a kitchen play dataset show that NILS can autonomously annotate diverse robot demonstrations of unlabeled and unstructured datasets, while alleviating several shortcomings of crowdsourced human annotations, such as low data quality and diversity. We use NILS to label over 115k trajectories obtained from over 430 hours of robot data. We open-source our auto-labeling code and generated annotations on our website: http://robottasklabeling.github.io.

**Keywords:** Language-Conditioned Imitation Learning, Data Labeling

## 1   Introduction

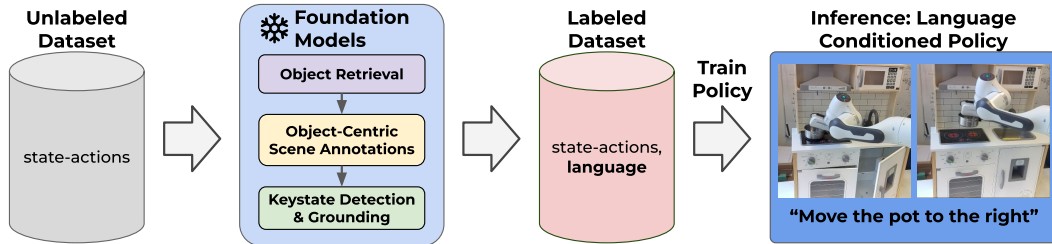

Figure 1: **A framework to label long-horizon robot demonstrations without human annotations or model training from RGB videos.** NILS leverages an ensemble of frozen pretrained models to segment and annotate uncurated, long-horizon demonstrations. The resulting labeled and segmented dataset can be used to train language-conditioned policies without human annotation.

Natural language is an intuitive and flexible interface for humans to communicate tasks to robots. Recent works have shown promising results in training language-conditioned policies using large

---

Correspondence to: nils.blank@kit.edu

8th Conference on Robot Learning (CoRL 2024), Munich, Germany.

datasets of robot trajectories paired with language annotations. However, the performance and diversity of behaviors learned by language-conditioned policies hinge critically on the quality and quantity of the language annotations. Unfortunately, most available robot interaction datasets do not contain any associated language annotations due to the costly nature of generating these. Concretely, most labeled datasets often rely on post-hoc crowd-sourced language labeling to generate labeled data [1, 2, 3, 4], which has several limitations: high cost in terms of time and money, variable quality of generated labels, reduced diversity when using templated instructions, and inconsistent granularity of annotations. The **quality** of the generated labels can vary significantly, from detailed descriptions to url links[1]. This issue can be addressed through templated language instructions [5], which in turn reduces the the **diversity** of the annotations, i.e., the same phrasing and instruction is used for every instance of the same skill, object or spatial relation. The least apparent downside of manually labeled behavior is the **granularity** of the annotations. The behavior description along the temporal axis can vary significantly from individual actions to whole multi-step tasks, e.g. `move left` and `clean the kitchen`, hence directly affecting the learned skills. This raises the question, can we find a more scalable and cost-effective way to generate semantically meaningful, detailed, and adjustable-granularity language annotations for existing robot interaction datasets?

One option is leveraging the semantic knowledge of Vision-Language Models (VLMs) pretrained on Internet-scale data [6, 7, 8]. These foundation models have found applications in different robotic contexts [9, 10, 11, 12, 13, 14, 15, 16, 17, 18, 19]. However, none of the existing models can accurately label robot demonstrations. Current VLMs struggle segmenting long-horizon demonstrations and tempo-spatial reasoning, especially in challenging robotic domains [17, 20].

To address these limitations, we introduce Natural language Instruction Labeling for Scalability (NILS), the first comprehensive system for zero-shot labeling of long-horizon robot videos without human intervention or additional model training. NILS employs an ensemble of pretrained foundation models to identify relevant objects and generate potential task labels. NILS first identifies object-centric keystates, segmenting long videos with multiple tasks into smaller single-action windows. Next, a Large Language Model (LLM) generates free-form language annotations based on structured observation descriptions. These descriptions are based on templated language generated from scene changes tracked by the pretrained foundation models. As a result, NILS converts videos of long-horizon robot interaction data into segmented and annotated datasets, which can be utilized for training language-conditioned policies without manual labeling. Furthermore, NILS addresses all of the above shortcomings, i.e., **cost**, **quality**, **diversity**, and **granularity**.

We demonstrate through extensive experiments, that NILS efficiently annotates unlabeled robot interaction data with appropriate task descriptions, surpassing state-of-the-art closed-source VLMs like Gemini-Pro. NILS reliably finds important keystates in long-horizon demonstrations better than prior zero-shot methods [21]. We demonstrate the scalability of NILS by labeling over 115k trajectories from different datasets, totaling 430 hours of robot data. Finally, we demonstrate the effectiveness of our approach by training language-conditioned manipulation policies on the automatically labeled data on a real robot.

## 2 Method

NILS consists of three stages: *Stage 1* (Identifying Objects in the Scene), *Stage 2* (Object-Centric Scene Annotation), and *Stage 3* (Keystate Detection and Label Generation). Figure 2 depicts a comprehensive method overview. The subsequent sections elaborate on each stage. NILS uses different frozen pretrained models across each state, which enable **modular replacement**. A detailed list of the models used in NILS, including explanations and usage, is provided in Appendix A.

---

[1]BridgeV2 dataset contains this annotation: "https://www.youtube.com/watch?v=JWA5hJl4Dv0"

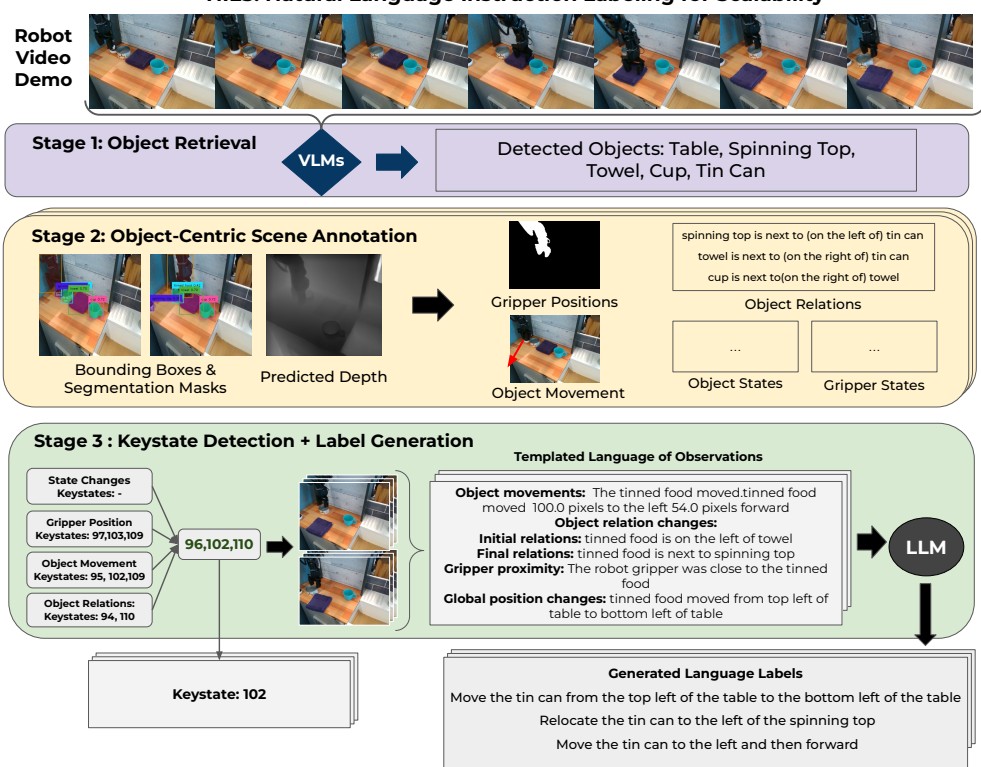

Figure 2: Overview of the proposed NILS framework for labeling long-horizon robot play sequences in a zero-shot manner using an ensemble of pretrained expert models. NILS consists of three Stages: First, all relevant objects in the video are detected. In the second step, object-centric changes are detected and collected. In Stage 3 the object change information is used to detect keystates and an LLM is prompted to generate a language label for the task.

## 2.1 Stage 1: Identifying Objects in the Scene

In Stage 1, NILS deploys a set of VLMs to detect all objects and their relevant properties across multiple video frames. This ensures comprehensive object detection even with occlusion. NILS prompts the model to output concise and unique names for all objects in each frame. However, this approach results in inconsistent naming across frames for the same object. To resolve this, NILS leverages *temporal consensus*, *co-occurrence*, and *object detection alignment*. It uses GroundingDINO [22] for bounding box generation and computes Intersection over Union (IOU) for co-occurrence. NILS combines the detector confidence with a SigLIP alignment score based on cropped image regions of the object. Finally, an LLM assigns the properties movable, is_container, states, and interactable to each object to help filter invalid robot-object interactions in later Stages.

## 2.2 Stage 2: Object-Centric Scene Annotations

Next, NILS generates object-centric scene annotations that allow to reason about robot-object interactions. For each object detected in Stage 1, NILS computes bounding boxes, segmentation masks, movement, relations, and state throughout the video. The extracted information is used in Stage 3 to segment and annotate the demonstrations. First, NILS identifies the object bounding boxes and segmentation masks to track various object changes.

**Object Annotations and Segmentations.** NILS computes bounding boxes and segmentation masks for the robot manipulator and all objects from Stage 1 using an ensemble of open-vocabulary detection and segmentation models. To address the detector confidence misalignment and class struggles,

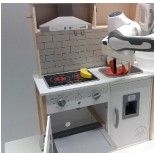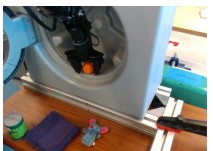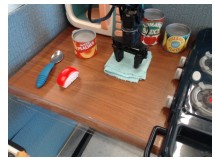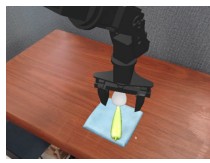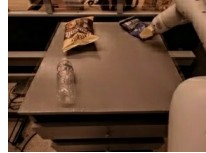

Figure 3: Overview of the environments used in our experiments. From left to right: Toy kitchen setup, two scenes from the BridgeV2 dataset [1], one example task from the Simpler Eval [26] and one scene from Fractal [5].

NILS ensembles the models by extracting bounding boxes for objects from Stage 1 (Subsection 2.1). Next, it computes the agreement between the detector and dense predictor inside each bounding box together with a temporal consensus approach. A two-stage filtering process identifies the most representative bounding box to improve the detection robustness [23]. To address temporal misalignments and missing detections, NILS utilizes a mask-tracking model [24] to capture temporal correlations between objects. Detailed explanations are provided in Appendix A.

NILS monitors four main signals: (1) object relations and movement using segmentation masks, bounding boxes, and depth maps, (2) object state changes over time, (3) gripper position relative to objects, and (4) gripper closing and opening actions. These signals are used to generate templated language descriptions and identify key states throughout the video, which are then used to describe the overall task. Detailed descriptions on how we track individual signals are provided in Subsection 2.4.

## 2.3 Stage 3: Keystate Detection and Label Generation

Determining critical states that mark task boundaries is a key challenge in labeling long-horizon robot demonstrations. NILS addresses this through a novel heuristic consensus method, using object-centric representations from Stage 2 to detect individual manipulation tasks. This method combines multiple heuristics to filter noise-induced false positives and identify reliable keystates. NILS acquires object-centric keystates by having each heuristic monitor changes for objects to minimize the noise impact. An object-centric keystate $o_i$ is considered valid if its score exceeds a threshold $\theta_o$

$$S(o_i) = \sum_k \alpha_k * S_k(o_i), \tag{1}$$

where $k$ is the heuristic index, $\alpha_k$ is the heuristic weight, and $Sk(o_i)$ is the confidence of the current heuristic. NILS uses equal weights for all heuristics. The keystate score $S_k(o_i)$ controls the **quality** of generated keystates and language annotations (Figure 10), providing an advantage over VLM captioning [25]. NILS then aggregates nearby keystates and selects the highest-scoring one to minimize noise.

**Action Retrieval and Grounding** After identifying keystates, NILS generates natural language descriptions of performed tasks. It constructs templated language prompts based on the object-centric scene annotations in between detected keystates. This information is used to query a LLM. The prompt focuses on a single object to help the LLM to concentrate on task-relevant information only. The LLM reasons about detected object movements and relation changes to determine possible robot actions. NILS detected low-level keystates, and templated language observations can thus be aggregated. Combining multiple low-level language observations in a single prompt allows the LLM to reason about higher-level tasks, allowing control over the task **granularity**. We provide detailed explanations and examples in Subsection B.2. A list of example prompts is given in Appendix D.

## 2.4 Object-Centric Information Retrieval of Stage 2

We provide further details on the object-centric information retrieval in Stage 2. Using bounding boxes and the segmentation masks, NILS monitors various signals for labeling. NILS tracks the following signals to generate templated language with timestamps:

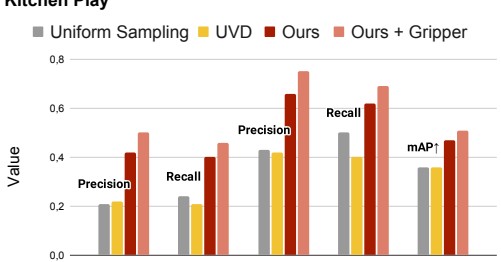
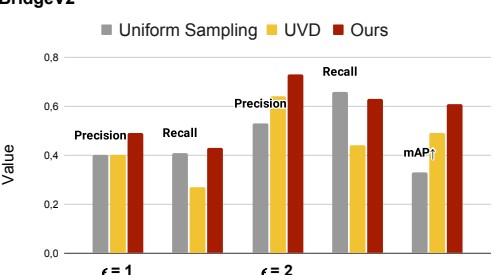

Figure 4: Keystate accuracy for different frame distance tolerances on: Kitchen Play and BridgeV2. We report the precision and recall of our method at two different keystate thresholds. NILS generates relevant keystates on both datasets and surpasses both baselines.

**Object Relations and Object Movement.** NILS tracks object movement and relations using segmentation masks and bounding boxes. It detects movement through bounding box displacement and flow-based detection [27] for small movements. NILS constructs an object-relation graph with nodes representing objects and edges denoting spatial relations [28]. To capture depth-dependent relations (e.g., `inside, behind`), objects are projected onto a point cloud using a zero-shot depth map [29]. NILS generates templated language describing movement and spatial relations, e.g., `"[] moved to the left of []"`. NILS generates **diverse** and informative labels based on these grounded scene observations, outperforming simple language-based paraphrasing methods.

**Object State Prediction.** NILS predicts object states over time, which is crucial for static objects like drawers. It crops object images from each frame and compares state text embeddings with image embeddings to determine the current object state. To handle occlusion during robot interactions, NILS omits predictions when relevant objects are partially obscured. The system generates templated language like `"Drawer changed from open to closed."`.

**Gripper Position.** The gripper position over time indicates robot-object interactions. NILS computes gripper-object proximity using object and robot segmentation masks and predicted depth map. To account for end-effector position inaccuracies, a per-object threshold is used to determine contact. A keystate is generated if the gripper-object distance is below the threshold for three frames with templated: `"The gripper was close to ..."`.

**Gripper Closing.** When available, NILS uses gripper close signals as potential keystates. NILS identifies a keystate when a previously closed gripper opens.

NILS collects these object changes and templated language observations throughout the video.

## 3   Evaluation

In this section, we address the central question: *Can NILS generate semantically meaningful, detailed, and adjustable-granularity language annotations for existing robot interaction datasets?* To answer this question, we assess NILS' accuracy in labeling long-horizon robot data, keystate quality, and grounding performance against state-of-the-art VLMs and study the generated task labels with respect to various properties. NILS was evaluated on three datasets: (1) the BridgeV2 [1] containing 1200 long-horizon trajectories with 28K short horizon pick and place and state-manipulation tasks with diverse scenes, tasks, and objects; (2) Fractal [5], a dataset consisting of 87K short horizon demonstrations; (3) a self-collected one-hour long-horizon, uncurated play dataset in a robot play kitchen, containing 439 short-horizon demonstrations of 12 tasks. The dataset consists of multiple long-horizon demonstration videos where the robot solves at least 10 manipulation tasks in a row.

## 3.1 Keystate Evaluation

We quantitatively evaluate the keystates produced by NILS on both datasets using precision, recall, and mean-average precision (mAP) metrics. A predicted keystate is considered correct if its distance to a ground truth keystate is smaller than a threshold based on the average short-horizon task length. Figure 4 shows that our framework outperforms UVD [21] and Uniform Sampling on the Kitchen Play and BridgeV2 at a keystate threshold of $\theta_o = 0.25$. Incorporating gripper-close signals further improves performance, highlighting the importance of including proprioceptive information for detecting keystates, if available. These experiments show NILS can extract valuable keystates in challenging environments.

## 3.2 Grounding Evaluation

We evaluate the grounding capabilities of NILS on a mixture of the BridgeV2 dataset and our Kitchen Play dataset. Specifically, we select 14 different long-horizon play demonstrations from various settings, each containing 20 tasks. Of these, 12 trajectories are sampled from BridgeV2. We chose trajectories from diverse settings, including kitchens, tables, and washing machines, with a wide variety of objects to ensure maximal task diversity. Further, we added two long-horizon demonstrations from our Kitchen Play Dataset with 24 tasks each. Given the ambiguity of language annotations, we employ human evaluators to assess whether a language label proposed by a method is correct given the segmented task video.

**Baselines.** We compare against four baselines. First, **Gemini Vision Pro** and **GPT-4o**, two large vision language models with video-understanding capabilities [30]. The prompt used to query these models is provided in Appendix D. We further test against **XCLIP**, a video-language retrieval model [31] and **VideoLLava**, a recent open-source VLM [32]. Since VLMs can not segment long-horizon demonstrations out-of-the-box, we provide the baselines with ground-truth keystates to assess their grounding capabilities.

**Results.** NILS outperforms all baselines in both settings (Table 1). The baselines struggle to ground the robot's actions due to the domain shift between the challenging static camera robot environment and pretraining data, consistent with previous findings [17, 33, 34, 35]. GPT-4o is the best-performing baseline, highlighting the performance difference compared to open-source VLMs. We found that the VLMs often correctly capture the interacted object but fail to perform correct temporal reasoning. Additionally, we observed that VLMs tend to successfully predict high-level state changes, such as opening or closing objects, but struggle with more nuanced interactions.

| Method | GT Keystates | Accuracy |
|---|---|---|
| XLIP | ✓ | 0.05 |
| VideoLLava | ✓ | 0.03 |
| Gemini 1.5 | ✓ | 0.16 |
| GPT-4o | ✓ | 0.51 |
| NILS | x | *0.66* |
| NILS | ✓ | **0.84** |
| NILS - TA | x | 0.46 |
| NILS - MFOR | x | 0.38 |
| NILS - DI | x | 0.47 |

Table 1: Grounding accuracy of NILS on our mixed dataset compared against baselines and ablations. NILS outperforms all baselines by a wide margin. TA: Temporal Aggregation, MFOR: Multi Frame Object Retrieval, DI: Depth Information. GT: Ground Truth Key States.

## 3.3 Labeling Results for BridgeV2 Dataset

To assess the scalability of NILS, we applied it to a larger, more diverse dataset. Therefore, we label a subset of BridgeV2 that consists of $31K$ trajectories in diverse environments. The cost for NILS to label this subset was approximately 200 USD, compared to about 5000 USD for crowd-sourced labels[2]. This demonstrates the **low-cost** of NILS. To analyze the diversity of generated labels, we visualized the top 60 labels, sorted by frequency, in Subsection B.1. This visualization confirms the **diversity** of labels generated by NILS compared to crowd-sourced annotations. Additionally, our

---

[2]Based on the reported labeling cost of approximately $10,000$ USD for the full dataset.

ablation experiments in Subsection B.2 showcase NILS's ability to generate task labels at various **granularities**, further highlighting its flexibility.

## 3.4 Language-conditioned Policy Training

| Method | Simulation | | | Bridge - Real Robot | | | | | |
| | Bridge | | Fractal | Spoon on Towel | | Sushi in Bowl | | Average | |
| | Success | CG | Success | Success | CG | Success | CG | Success | CG |
|---|---|---|---|---|---|---|---|---|---|
| Gemini | 0% | 11% | - | - | - | - | - | - | - |
| GT | **4.2%** | **41.7%** | 1.9% | 60% | 93% | 60% | 73% | 60% | 83% |
| NILS | 2.8% | 29.6% | **3.1%** | **66%** | **100%** | 60% | **80%** | **63%** | **90%** |

Table 2: Success rates for simulated (left) and real environments (right) across different tasks. We evaluate all methods on grounding ability (CG) and task success rate. Results are averaged over 15 rollouts for each task in the real robot environments.

To study the generalization of NILS for large-scale, diverse datasets, we train a language-conditioned policy with labels from NILS on the BridgeV2 [1], and Fractal [5] datasets. We use Octo [36], a recent open-source diffusion policy and train it using 3 different label sets: crowd-sourced labels (GT), NILS labels, and Gemini labels. We train Octo [36] on different data mixtures to evaluate the capabilities of our labeling framework for downstream policy learning. In particular, we train Octo on the subset of BridgeV2 labeled by NILS and Fractal fully labeled by NILS. We always train the baseline policies with the same datasets, but switch out the language labels with their respective counterparts. Due to labeling costs and Gemini's low performance, we omit labeling Fractal with Gemini. Further evaluation details are provided in Appendix F.

**Results.** The results of this experiment are summarized in Table 2. In the simulated environment, our method outperforms the policy trained on labels generated by Gemini by 19% in terms of correct grounding and 2.8% in terms of success rate while being slightly behind the crowd-sourced anno-tated dataset for BridgeV2. On Fractal, the policy trained with NILS labels outperforms the policy trained on ground-truth labels by 1.1%. For the real robot tasks, the policy trained with labels from NILS achieves the highest average performance with a 63% success rate. These results align with our findings in the previous section on label accuracy. Our experiments demonstrate that current VLMs alone are insufficient to produce high-quality labels for effective policy training. Further-more, these results highlight that NILS generates labels of **high quality** even for diverse, large-scale datasets, making it a promising approach for scalable robot learning.

## 3.5 Ablation Studies

**What are the most crucial components of NILS?**

We conduct ablation studies to investigate the importance of various design decisions in NILS. Results are summarized in the lower half of Table 1. Without multi-frame object retrieval in Stage 1, NILS achieves only half the grounding accuracy, highlighting the importance of accurate and complete initial object labels. Our experiments also indicate that the depth in-formation provided by the zero-shot depth prediction model (DI) in Stage 2 is crucial for accurate spatial reasoning, en-abling more precise task labeling. Given the current limita-tions of foundation models, we found that temporal aggre-gation (TA) is essential for obtaining reliable and consistent bounding boxes and segmentation masks across frames. Thor-

| Method | Captured Objects |
|---|---|
| Ours (Gemini) | **0.94** |
| Ours (Gemini) + SOM | 0.74 |
| Ours (GPT4V) | 0.67 |
| Ours (Prismatic [35]) | 0.48 |
| Single Frame (Gemini) | 0.70 |
| OWLv2 + SigLIP | 0.63 |

Table 3: Ablation of the effective-ness of our initial object retrieval measured in recall.

ough ablations for every design decision in all Stages of NILS are provided in Appendix C, confirm-ing the importance of each Stage for achieving accurate labels.

**How to retrieve all relevant objects in the scene?** To reason about robot-object interactions, it is crucial to detect all relevant objects in the scene. We investigated various VLMs and combina-

tions for object retrieval in Stage 1. We compare NILS design using Gemini with *co-occurence* and *detection alignment* against several variants using other VLMs like GPT-4V and open-source baseline Prismatic [35]. Additionally, we tested a combination of popular open-source open-vocabulary models: SigLip [7] + OwlV2 [37]. To assess the effectiveness of all approaches, we create a diverse dataset consisting of the BridgeV2 and our Kitchen Play dataset. In total, the ablation dataset comprises 185 objects in various scenes. The results for this experiment are summarized in Table 3. Naively querying a VLM for a single frame does not reliably capture all objects, especially in cluttered scenes. NILS without co-occurrence and detection alignment, denoted as "Single Frame", only captures 70% of the objects present in the scenes, emphasizing the importance of filtering and improving initial object detections. While OWLv2+SigLIP presents a viable alternative, it requires prior knowledge about objects that might appear in the scene. We also ablate Set of Mark Prompts (SOM) [38], with our method, but found that this harms performance.

## 4    Related Work

**Key State Identification.**  Identifying important keystates from long-horizon tasks is crucial for efficiently learning goal-conditioned policies.  Some approaches use proprioceptive observations [39, 40, 41, 42] or waypoint reconstruction loss [39]. UVD utilizes the latent space of pretrained image embedding models such as CLIP [21] to detect keystates. Similar to NILS, it also does not require any robot signals. However, NILS additionally incorporates more specific information, which significantly improves keystate quality. REFLECT [28] constructs a scene graph with object relations and states, labeling frames as keyframes when the graph changes. However, this method relies on ground truth state information and object positions, which are usually only available in simulated environments. NILS detects keystates in real-world environments from robot videos only.

**Action Recognition.**  Video action recognition involves retrieving actions performed over multiple frames, either through dense video action recognition (extracting multiple actions and their time frames) or video action classification (assuming a single action per video). Generalist VLMs can solve these tasks in a zero-shot, open-vocabulary manner [43, 44, 45, 46, 47]. RoboVQA [17] finetunes a video VLM on a labeled robot demonstration VQA dataset to answer questions about robot actions in robotics. REFLECT [28] extracts scene graphs that could be used for action retrieval but assume a known robot plan to analyze robotic failure cases. Several works fine-tune CLIP [48, 49, 50, 51] using a small subset of domain-specific data. These CLIP models then retrieve actions for a larger, unlabeled dataset. These approaches assume known key states and require labeled in-domain finetuning data. To the best of our knowledge, NILS is the only method that accurately labels real-world long-horizon robotic data without any model fine-tuning.

## 5    Discussion

**Limitations.**  Despite NILS' ability to generate high-quality labels, some limitations remain: Using multiple models for generating scene representations results in a significant computational cost, requiring 7 minutes to label one 8-minute long-horizon trajectory consisting of 50 tasks on a 3090 RTX GPU. Correlated heuristics can sometimes lead to high-confidence keystates triggered by noise, resulting in incorrect labels. Grounding accuracy is limited by the performance of pretrained models. NILS has high objectness assumptions for labeling, making it challenging to label tasks with granular objects and distinguish between visually similar objects due to the limitations of object detectors.

**Conclusion.**  This work introduces NILS, the first framework to autonomously label long-horizon robot datasets without human intervention or any model training. NILS offers a **cost-effective** alternative to annotate robot demonstrations compared to crowd-sourcing. Our experiments and extensive ablations showcase NILS' capability to generate labels of **high quality** and **diversity**, enabling efficient and scalable language-conditioned robot learning.

## Acknowledgments

The work presented here was funded by the German Research Foundation (DFG) – 448648559. The authors acknowledge support by the state of Baden-Württemberg through HoreKa supercomputer funded by the Ministry of Science, Research and the Arts Baden-Württemberg and by the German Federal Ministry of Education and Research.

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

# A  Additional Method Details

## A.1  Overview of Pretrained Models for NILS

We summarize all used pretrained foundation models for NILS. All models are used without any fine-tuning. We want to highlight that all models can be exchanged for other pretrained models of the same category. Further, an overview of the usage of all different models is provided in Table 4.

| Model Class | Stage1 | Stage2 | Stage3 |
|---|---|---|---|
| VLM | ✓ | | |
| Contrastive Vision-Language Encoder | ✓ | ✓ | |
| Open Vocab. Detec. 1 | ✓ | ✓ | |
| Open Vocab. Detec. 2 | ✓ | | |
| Mask Tracking | | ✓ | |
| Bounding Box Clustering | | ✓ | |
| Depth Estimation | | ✓ | |
| Large-Language Model | ✓ | | ✓ |
| Flow Estimation Model | | ✓ | |

Table 4: Summary of each pretrained model usage across each Stage of the NILS framework.

**VLMs for Object Retrieval: Gemini V 1.5 [30]** NILS detects all objects in a scene by prompting a VLM for multiple frames. We found that Gemini 1.5 surpasses other VLMs for initial object retrieval, as shown in Table 3.

**Contrastive Vision-Language Encoder: SigLiP [7]** NILS uses SigLiP as our Contrastive Vision-Language Model (CVLM) due to it showing strong performance across many benchmarks. Given an image and several text descriptions, SigLip computes a score that measures the compatibility of text and images. NILS uses SigLiP in Stage 1 to improve the object detection accuracy when multiple labels are available for the same object. In Stage 2, SigLip is used to track object changes of static objects, such as a drawer, by cropping the image part of the scene with the relevant object and computing scores for all possible object changes.

**Open Vocabulary Detection 1: GroundingDINO [22]**

NILS uses GroundingDINO as the main open-vocabulary detector for object detection. We found that GroundingDINO is better at detecting more specific object descriptions containing colors, while OWLv2 is better at detecting simple object descriptions. This can be attributed to the token-matching loss of GroundingDINO. As the initial object list is generated by a generative VLM, the object descriptions tend to be more specific, and OWLv2 often fails to capture these relations, so we opt for GroundingDINO.

**Class-Agnostic Bounding Boxes: OWLv2 [37]** During the initial object retrieval stage, NILS uses class-agnostic bounding boxes and their corresponding objectness scores from OWLv2 to filter out low-objectness objects. IOU matching is employed to associate the class-agnostic bounding boxes with the objects detected by the VLM.

**Mask Tracking: DEVA [24]** DEVA is a mask tracking framework based on XMem [52], which merges new incoming detections with propagated detections for temporal consistent masks. We utilize DEVA with a few simple extensions to obtain temporally robust masks. This is especially important for cases in occlusion, where the object detector would falsely detect another object if the actual object is not visible.

**Bounding Box Clustering: DBSCAN [23]** Often, static objects can have different states. To ensure robust state predictions for static objects, such as a drawer or an oven, NILS performs static object bounding box refinement as described in Sec. A.3

**Depth Estimation: DepthAnything [29]** Depth is an important metric for capturing spatial relationships between objects. Since NILS labels demonstrations from RGB videos that do not contain

depth information, we approximate the depth using DepthAnything [29] as an approximation. NILS uses the model in Stage 2 to monitor relevant object relations when the robot manipulates objects and changes in their positions.

**Large-Language Model: Gemini Pro** In Stage 1, NILS uses an LLM to generate possible object states, and in Stage 3, it employs the LLM to generate language labels for the short sequence. NILS utilizes Gemini Pro for both tasks due to its integration with Gemini V, which is used as a VLM in the earlier stages, and its lower costs compared to GPT-4.

**Flow Estimation Model: GMFLOW [27]** To detect object movements that do not show bounding box displacement, such as cabinet opening, NILS incorporates optical flow information in addition to the object's bounding box.

## A.2 Stage 1: Identifying Objects in the Scene

Initially, NILS utilizes 8 equally distributed frames from a long-horizon demonstration video to query Gemini. For each frame, NILS prompts the model to output concise, specific, and unique names for all objects in the scene, as well as their colors. Figure 16 in the Appendix shows the prompt used for initial object retrieval. Querying the VLM for multiple frames significantly increases the number of detected objects, particularly in cases of occlusion. However, the same object is now likely referred to under different names in different frames. Thus, NILS leverages *temporal consensus*, *co-occurrence*, and *object detection alignment* to select consistent and representative object names across frames. For temporal consensus, NILS prompts GroundingDINO with all object descriptions generated by the VLM for all 8 frames to generate a set of bounding boxes for each frame. Next, co-occurrence is measured for each individual frame by computing bounding box IOUs, capturing different names referring to the same object. If the IOU is above a certain threshold, NILS assumse that the corresponding object names refer the same object. The co-occurring objects are counted and grouped for all 8 frames, and a representative object name is chosen according to object detector confidence. Object names with the highest confidence scores per group are stored in synonym lists to enrich the instruction. This approach generates contextualized grounded object synonyms, increasing the overall data diversity. Furthermore, this approach can be considered as automatic prompt engineering. Some object descriptions can result in better, more consistent bounding boxes produced by the object detector. By selecting the object name with the highest confidence, we also select the object name that is most likely to produce correct detections for the other frames. Finally, an LLM assigns the properties movable, is_container, states, and interactable to each detected object. These properties help to filter wrong robot-object interactions in subsequent Stages.

As a simple baseline, we employ a combination of OWLv2 [37] and SigLIP [7]. Given a predefined list of objects commonly appearing in robot environments, we compare class-agnostic bounding boxes generated by OWLv2 with class embeddings of all objects in the list. We further cut out objects based on their class-agnostic bounding boxes and compare text-image similarity with SigLIP. Finally, we average the scores of OWLv2 and SigLIP for each class-agnostic bounding box and retrieve the object with the highest score. This approach is promising when we have a pretrained list of objects but falls short for similar objects, such as different colored objects of the same instance.

For set-of-mark prompting, we annotate the frames with enumerated class-agnostic bounding boxes generated by OWLv2 and task the VLM to assign an object name to each numbered bounding box.

## A.3 Stage 2: Object-Centric Scene Annotations and Information Retrieval

**Object Filtering and Mask Refinement through Temporal Aggregation.**

Detecting keystates accurately based on object masks and boxes requires temporally consistent object masks. However, initial object detections may suffer from temporal misalignments, such as missing detections for certain frames or an object being classified with a synonym for different frames. To address this challenge, NILS utilizes DEVA [24], a mask-tracking model, to capture temporal correlations between objects. DEVA propagates masks with X-Mem [52] while frequently

incorporating new segmentations. The new segmentations are fused with the propagated segmentations through alignment. NILS further extends DEVA to incorporate a class score for each propagated mask. The resulting final mask belonging to each object then has multiple class scores of possibly different classes associated with it. NILS obtains the most confident class and labels the object as the determined class. NILS extends DEVA to incorporate class scores for each propagated mask, obtaining the most confident class for each object. This results in bounding boxes for all timesteps and more robust predictions. NILS also applies techniques to mitigate DEVA issues, such as merging masks of the same class, keeping the highest intersection-over-union mask component, and filtering masks for temporally consistent and coherent class labels. In particular, DEVA masks often split if objects of similar visual features overlap. We mitigate this issue by computing individual mask components and removing the component with low area. After applying DEVA and filtering, the masks are temporally more consistent and have consistent class labels.

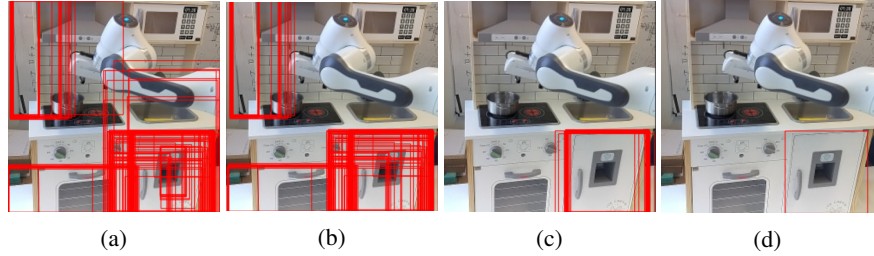

|     |     |     |     |
| :-: | :-: | :-: | :-: |
| (a) | (b) | (c) | (d) |

Figure 5: **Static object detection refinement.** (a) shows the initial noisy labels. The boxes are then filtered by removing statistical outliers (b) and by obtaining the highest confidence cluster (c). The final averaged box is visible in (d)

**Increasing Detection Robustness for Static Objects.** For non-moving objects, NILS employs a temporal consensus approach to enhance detection robustness and accuracy, especially in scenarios prone to occlusion. A two-Stage filtering process identifies the most representative bounding box for static objects over time, first by eliminating statistical outliers and then by clustering the remaining boxes using DBSCAN [23], which also detects noise labels. The final bounding box for each static object is derived from the cluster with the highest overall confidence, representing the object consistently across frames.

### A.3.1 Object-Centric Information Retrieval

**Object Relations and Object Movement** We generate an object-relation graph from segmentation masks generated in the first step. First, the objects are projected into a point cloud with a depth map generated by Depth-Anythingv2 [29]. We further perform additional filtering steps, such as outlier detection and downsampling. To reason about object movement in natural language in camera space, we project the pointcloud into a coordinate frame that aligns with flat surfaces in the scene. First, NILS detects surfaces, such as *floor, stove top, table, counter* in the scene. The detected surface is projected into the pointcloud, followed by plane segmentation and normal estimation. The normal acts as the upvector of the new coordinate system, whereas the front-vector points towards the camera. Furthermore, many tasks include changing the position of an object with respect to the surface it is located on. To detect the position of objects on a surface, NILS uses the previously detected surface object and performs a homography transform to account for camera perspective. To do this, we fit a quadrilateral to the surface's segmentation mask and compute a transformation matrix from the detected corners to image corners. We then project the objects bounding boxes with this matrix and compute object position on the surface by categorizing positions in a 3x3 grid. **Object State Prediction.** NILS crops objects from images based on their bounding box. We add a small padding to the bounding box, to ensure all relevant information is present in the cropped image. Robot mask IOU determines occlusion with the cropped region. Then, NILS compares the CLIP similarities of state text embeddings and the cropped images. The prompts have the form `"A picture of a <state> <object>"`.

# B Additional Experiments

## B.1 Diversity Comparison of Generated Task Labels

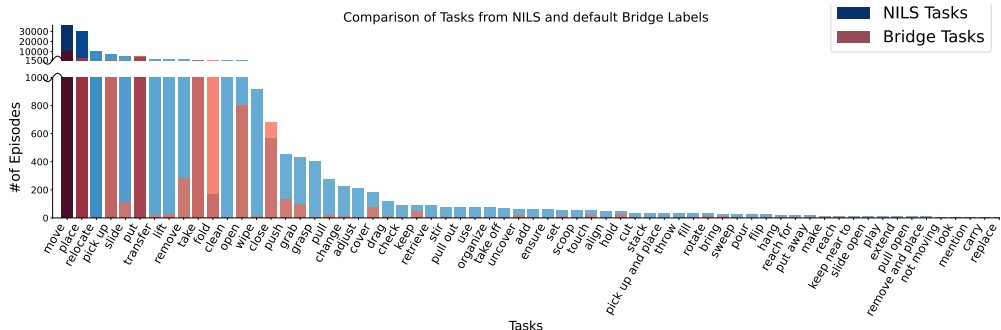

(a) Comparison of the top 60 tasks labels generated by NILS and the top 60 crowd-sourced annotation on the BridgeV2 dataset.

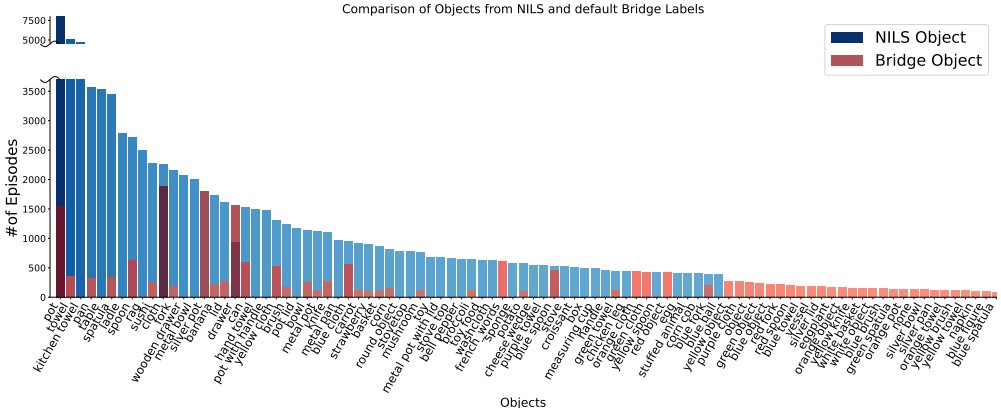

(b) Comparison of the top 60 object labels generated by NILS and the top 60 crowd-sourced annotation on the BridgeV2 dataset.

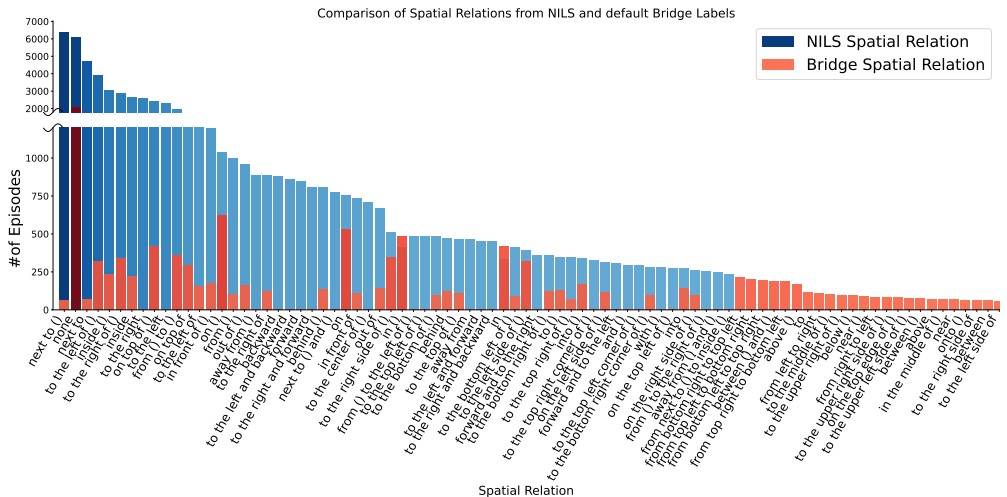

(c) Comparison of the top 60 spatial arrangement labels generated by NILS and the top 60 crowd-sourced annotation on the BridgeV2 dataset.

Figure 6: Comparison of labels generated by NILS and crowd-sourced annotation on the BridgeV2 dataset. The Bridge version used by NILS is around 4 times bigger then BridgeV2.

## B.2 Hierarchical Instruction Generation

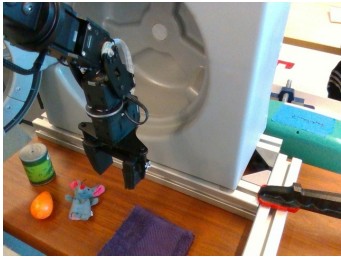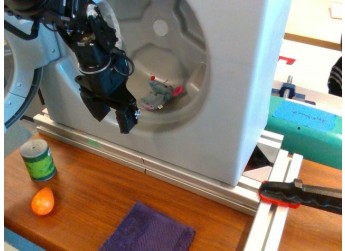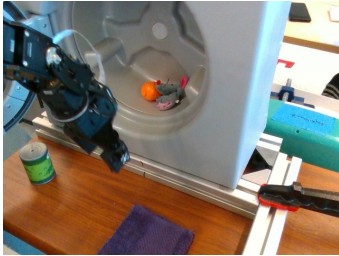

**Fine-Grained Tasks**: Move the toy mouse to the center of the washing machine - Place the plastic egg next to the toy mouse
**High-Level Tasks**: Put the toys in the washing machine.

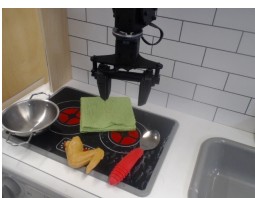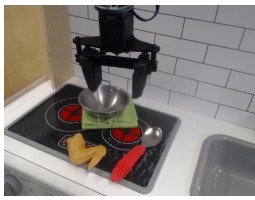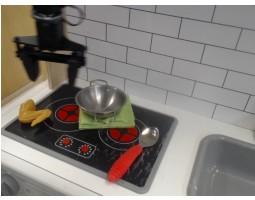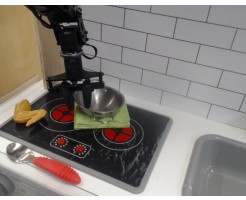

**Fine-Grained Tasks**: Reposition the pan on the kitchen towel - Pick up the chicken wing and place it on the left of the pan - Move the spoon to the bottom left of the stove top
**High-Level Task**: Prepare the stove top for cooking the chicken wing.

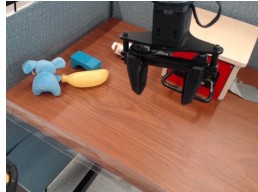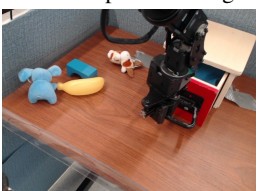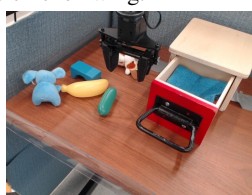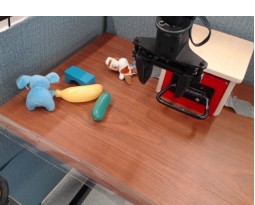

**Fine-Grained Tasks**: Open the red drawer of the wooden box - Move the green pickle toy from the wooden box to the center of the table - Close the red drawer of the wooden box
**High-Level Task**: Fetch the green pickle toy from the box and place it with the other toys.

Figure 7: Examples of different levels of tasks granularity generated by NILS for two trajectories from the Bridge dataset.

NILS produces keystates and language annotations, which are contextually grounded, for single tasks. By combining the templated language descriptions of consequent keystates into a single prompt, the framework can generate high-level language instructions. This allows for granularity control over the produced tasks, which can be beneficial for policy learning. [53]. We illustrate three examples from the Bridge Dataset in Figure 7.

## C  Ablations

### C.1  Stage 1: Initial Object Retrieval Ablations

To reason about robot-object interactions, it is crucial to initially capture all objects in the scene reliably. Our initial object detection is based on a multiple-frame consensus that reliably works in different settings and for different objects. To assess the effectiveness of our approach, we create a diverse dataset consisting of the BridgeV2 and our Kitchen Play dataset. In total, the evaluation dataset comprises 185 objects. We compare our approach with different VLMs and two baselines. The first baseline queries a VLM based on a single frame. In OWLv2 + SigLIP, we utilize a predefined list of about 1400 objects commonly appearing in kitchen environments. Then, we generate

class-agnostic bounding boxes with OWLv2 and align all class embeddings with the class-agnostic box embeddings generated by OWLv2. Furthermore, an ensembling approach combines the OWLv2 scores with cropped image-text alignment scores generated by SigLIP. Figure 9 shows examples of multi-frame retrieval outperforming naive, single-frame retrieval.

## C.2 Stage 2: Scene Annotation

| | | ($\epsilon = 8$) | | ($\epsilon = 16$) | | |
|---|---|---|---|---|---|---|
| | | Amb. | Single | Amb. | Single | |
| | Naive | 0.59 | 0.34 | 0.60 | 0.33 | |
| Grounding | Naive - SG | 0.62 | 0.27 | 0.59 | 0.26 | |
| | - Temporal Alignment | 0.76 | 0.53 | 0.68 | 0.51 | |
| | - Detection ensembling | 0.59 | 0.50 | 0.59 | 0.50 | |
| | F.F. | 0.80 | 0.61 | 0.75 | 0.57 | |
| | | Precision | Recall | Precision | Recall | mAP ↑ |
| | Naive | 0.46 | 0.36 | 0.67 | 0.53 | 0.36 |
| Keystates | - Temporal alignment | 0.41 | 0.45 | 0.70 | 0.77 | 0.45 |
| | - Detection ensembling | 0.46 | 0.48 | 0.69 | 0.73 | 0.48 |
| | F.F. | 0.50 | 0.46 | 0.75 | 0.69 | 0.51 |

Table 5: Ablation for the effectiveness of our perception filtering on our Kitchen Play Dataset. For Naive, we simply use OWL-v2 and SAM to extract masks and bounding boxes without additional filtering or temporal aggregation. In Naive-SG, we provide a full object-relation prompt to the LLM instead of an object-centric prompt when retrieving the action. F.F. depicts full filtering.

Detecting all relevant objects in the scene in Stage1 is crucial for the performance of NILS. In Table 5, we provide ablations of our perception module. To assess the effectiveness of our perception module, we compare against simple box generation with OWLv2 and object segmentation with Efficient-SAM [54]. We perform ablations by disabling several components: ensembling with a dense open vocabulary predictor, statistical mask outlier filtering, temporal aggregation, state prediction without occlusion, and static object box aggregation. We conduct all experiments with a keystate threshold of $0.3$ and Gemini as the LLM. When we omit our heavy postprocessing Stages, we observe a significant decline in keystate quality and grounding accuracy. Although the drop in keystate precision is not substantial, the recall shows a notable decrease. Additionally, the grounding accuracy drops significantly, especially when only unambiguous prompts are considered valid. We observed that constraining the prompt information to a specific object and its relations helps to reduce hallucination and results in more precise predictions. These findings underscore the necessity of robust post-processing techniques to effectively leverage current state-of-the-art perception models in novel and challenging domains.

## C.3 Stage 3: Keystate Ablations

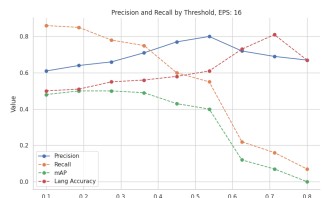

Figure 10: Keystate detection precision and recall for different threshold values for our Kitchen Play dataset.

Next, we analyze the importance of our keystate heuristic to determine critical states. Figure 8a shows the performance of our method when incorporating different keystate heuristics. Gripper close signals present a very strong baseline. However, as mentioned before, gripper close signals are not always available and can not represent all different kinds of tasks. This is shown by the increased precision and recall when incorporating additional heuristics. Especially for a smaller threshold, we observe a significantly increased performance when incorporating additional heuristics. Incorporating additional heuristics usually results in an increase in precision and a decrease in

| | | $\epsilon = 8$ | | $\epsilon = 16$ | |
|---|---|---|---|---|---|
| NILS | gripper close | 0.35 | 0.32 | 0.73 | 0.65 |
| Gripper | gripper close + object state | 0.38 | 0.36 | 0.73 | **0.69** |
| | all | **0.50** | **0.46** | **0.75** | **0.69** |
| | all | 0.42 | 0.40 | 0.66 | 0.62 |
| NILS | object relations | 0.21 | 0.13 | 0.56 | 0.35 |
| RGB | + gripper pos. | 0.29 | 0.40 | 0.52 | 0.71 |
| | + gripper pos. + state | 0.39 | 0.32 | 0.62 | 0.51 |
| | object movement | 0.31 | 0.40 | 0.52 | 0.68 |
| | + gripper pos. | 0.34 | 0.42 | 0.56 | 0.70 |
| | + gripper pos. + state | 0.42 | 0.38 | 0.65 | 0.59 |
| | gripper pos. | 0.31 | 0.43 | 0.50 | 0.69 |

(a) Keystate precision and recall when using different keystate heuristics. For all experiments, the keystate detection threshold is set to 0.3, if applicable.

recall. Always using all heuristics is desired, as the precision-recall tradeoff can then be best controlled by setting an appropriate threshold. Figure 10 depicts the relation between threshold, keystate precision and recall, and grounding accuracy. With increasing threshold, the grounding accuracy and keystate precision increase. This indicates that with our scoring method, the quality of samples can be controlled effectively.

### C.3.1 Stage 3: Simulation Keystate Ablation

| | | $\epsilon = 8$ | | $\epsilon = 16$ | |
|---|---|---|---|---|---|
| Method | | Precision | Recall | Precision | Recall |
| UVD | VIP | 0.16 | 0.06 | 0.28 | 0.10 |
| NILS | | **0.37** | **0.21** | **0.53** | **0.31** |

Table 6: Keystate accuracy for different frame distance tolerances on CALVIN Simulation Benchmark [2] with RGB Image-Data Only. We compare against UVD, which uses VIP [55] to detect keystates. NILS outperforms UVD in this experiment.

We further evaluate NILS on a low-resolution simulation environment that has a high domain shift compared to our real-world setting. For this experiment, we test our framework on the CALVIN simulation benchmark [2]. CALVIN is a challenging benchmark that has long-horizon play data. In Table 6 we show the keystate detection precision and recall on the CALVIN [2] benchmark. NILS utilizes off-the-shelf models trained on real-world data. Thus, these models struggle significantly in simulated environments that contain abstract objects. We had to perform prompt engineering to make the detection and OV-segmentation models detect any objects in the scene. After these changes, the method performs reasonably well.

### C.4 Stage 3: Choice of LLM for Label Generation

We further compare the annotations produced by NILS with ground truth language annotations obtained through human labeling on the Kitchen Play dataset. To allow for a quantitative analysis of the generated language annotations, we phrase the problem as a multiple-choice problem. Instead of generating a language instruction directly, we provide a list of possible tasks from which the LLM has to select up to two tasks, given the observations made in Stage 2. Table 7 shows the grounding accuracy of our framework compared to Gemini, S3D, and XCLIP. NILS outperforms all baselines by a large margin, whereas GPT-4 is the best LLM.

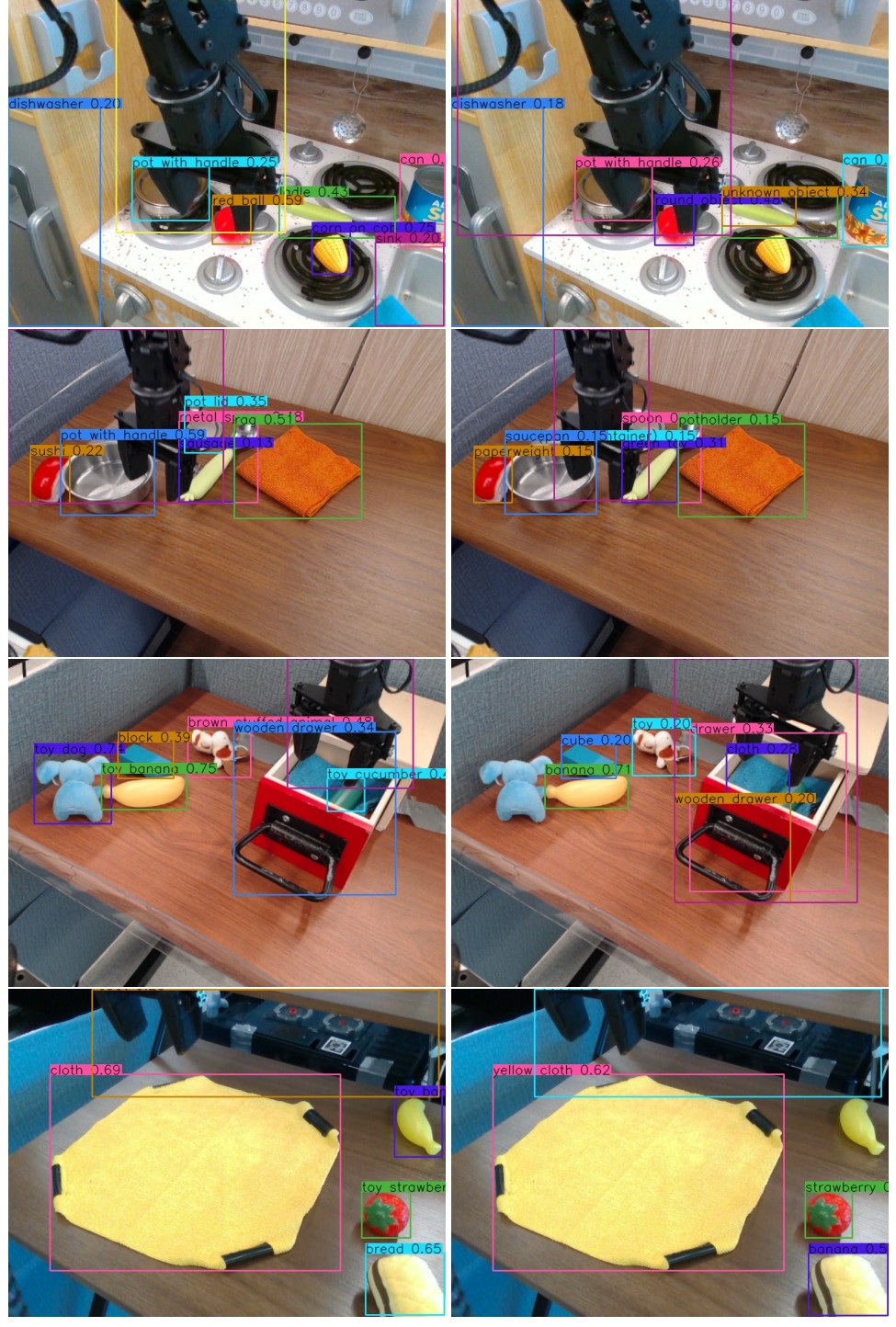

Multi-frame initial object retrieval      Single-frame initial object retrieval

Figure 9: Comparison of multi-frame vs single-frame object retrieval. The single-frame retrieval often misses objects, and the overall label quality is worse than those of the multi-frame aligned approach.

| Method | LLM | Accuracy ($\epsilon = 8$) | | Accuracy ($\epsilon = 16$) | |
|---|---|---|---|---|---|
| | | Amb. | Single | Amb. | Single |
| S3D | - | | 0.04 | | 0.03 |
| XCLIP | - | | 0.07 | | 0.09 |
| Gemini | | | 0.13 | | 0.13 |
| | GPT-3.5 | 0.70 | 0.55 | 0.67 | 0.53 |
| NILS | GPT-4 | **0.84** | **0.77** | **0.79** | **0.71** |
| | Gemini (Lang) | 0.80 | 0.61 | 0.75 | 0.57 |
| | Mixtral8x7b | 0.66 | 0.52 | 0.62 | 0.49 |

Table 7: Grounding accuracy of our framework on our Kitchen Play Dataset. For Amb., the prediction is labeled correct if the list of answers contains the ground truth. In Single, ambiguous predictions are wrong.

# D    Example Prompts

We give example prompts used to generate the list of potential tasks given a list of objects in Figure 17. In Figure 18, an example prompt used to label the task a robot solved in between two keystates is visualized.

# E    Qualitative Examples

We show qualitative examples of our framework's produced natural language instructions on BridgeV2 [1] in Figure 21.

# F    Additional Experiments

## F.1    Evaluation Details

To evaluate policy performance on the BridgeV2 dataset, we use both sim and real robot experiments using a 6-DoF WidowX robot arm. To evaluate perfomance on the Fractal robot, we rely on simulation only. Simulation experiments are conducted using the Simpler Environments [26], and real robot experiments are performed using scenes inspired by the Bridge V2 dataset [1]. One Simpler Task is also visualized in the right image of Figure 3. All Octo variants are trained for 300k steps on their respective label sets to ensure a fair comparison. In the real robot environment two tasks were tested: *Move the spoon inside the towel* and *Place the sushi inside the [color] bowl*, both with various distractors and different starting positions for each object. The tasks are visualized in Figure 11 in the Appendix. We report success rate and correct grounding, where correct grounding refers to the robot approaching and interacting with the task-relevant object.

## F.2    Bridge V2 Experiments

| Method | Spoon on Tablecloth | | Carrot on Plate | | Eggplant in Basket | | Average | |
|---|---|---|---|---|---|---|---|---|
| | CG | Full | CG | Full | CG | Full | CG | Full |
| NILS | 43.1% | 6.9% | 12.5% | 0% | 33.3% | 1.4% | 29.6% | 2.8% |
| Bridge Gemini Baseline | 18% | 0% | 16% | 0% | 0% | 0% | 11% | 0% |
| GT | 37.9% | 11.1% | 40.3% | 1.4% | 27.8% | 0% | 41.7% | 4.2% |

Table 8: Success rates for simulated environment methods across different tasks. CG (Correct Grounding) refers to the robot successfully grasping the task-relevant object.

We conduct the experiments for Bridge V2 in SIMPLER [26] and on a real robot. For the real robot setup, we perform 15 rollouts per task with different initial states and distractors. For simulation, we follow the setup provided in SIMPLER. The real-robot and simulation setups can be seen in Fig. 11. Table 8 shows partial and full success rates for three different tasks in the simulated environment. The simulation additionally provides a partial success metric, which requires the robot to grasp the task-relevant object. This metric can be interpreted similarly to our correct-grounding metric.

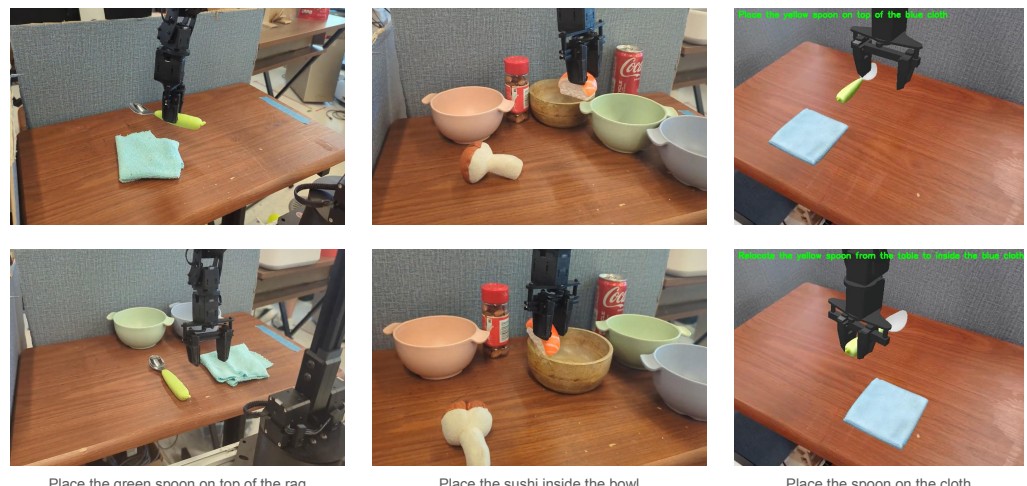

| Place the green spoon on top of the rag | Place the sushi inside the bowl | Place the spoon on the cloth |

Figure 11: Tasks used for evaluation in BridgeV2 in real-world (left) and simulation (right)

## F.3 Fractal Experiments

| Method | Open Drawer | | Close Drawer | | Move Near | | Pick Coke Can | | Average |
|--------|------|------|------|------|------|------|------|------|---------|
| | VM | VAR | VM | VAR | VM | VAR | VM | VAR | |
| NILS | 1.5% | 0.5% | 16.3% | 1.5% | 6.2% | 2.9% | 2% | 0.7% | 3.1 % |
| GT | 0% | 0% | 9.3% | 1.6% | 6.7% | 2.1% | 1% | 0% | 1.9% |

Table 9: Success rates for simulated environment methods across different tasks. VM refers to visual matching, where the surfaces and environments are similar to the environment in the training data. VAR refers to variant, where the scene is located in different rooms and the surfaces have different textures.

Furthermore, we conduct experiments with a dataset collected on a Google Robot [5]. We obtain the dataset from Open-X Embodiment [56] and label all 87k trajectories with NILS. We train Octo on the labeled dataset and additionally co-train a policy with BridgeV2 labeled by NILS, where we choose the dataset weights so each dataset appears equally often during training. We found that cotraining with BridgeV2 improves the performance on Fractal. We again evaluate the policies in SIMPLER [26], where the robot has to solve 3 tasks with several different variations. Table 9 shows the success rate for different tasks in SIMPLER. The policy trained with NILS outperforms the policy trained on crowd-sourced annotations.

### F.4 Real Robot Experiments in the Toy Kitchen

The following section describes our real-world play kitchen environment conducted in a toy kitchen environment in detail. We collect play data through teleoperation in our robot kitchen environment. The setup is illustrated in Figure 19. The robot can solve 12 different tasks, as shown in Figure 20.

We evaluate the performance of the trained policies based on two metrics: **Success Rate.** We perform each task three times and calculate the average number of successful task completions. We then compute the average success rate over all tasks. **Correct Grounding.** We evaluate whether the policy correctly understands language instructions. The task does not need to be completed successfully. The robot only has to show that it correctly understood the task. For instance, if the robot approaches the oven and tries to open it but fails, we label the task as correctly grounded. We again compute the average over all possible tasks.

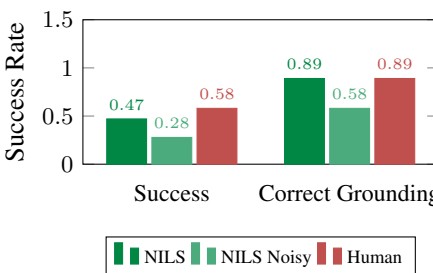

Figure 12: Policy performance trained on a play dataset with ground-truth labels, segmented trajectories and labels from NILS and noisy labels from NILS. The policy based on NILS' labels is competitive with the one trained on human-annotated labels. Incorporating low-quality labels hurts the policy's performance.

We train three policies: Human, a policy trained on ground truth, human-annotated data; NILS, a policy trained on data labeled by NILS, where we discard ambiguous labels; and NILS Noisy, where, if the LLM is unsure about the tasks and outputs multiple language instructions, we incorporate the demonstration in the training dataset multiple times with each generated instruction. The grounding accuracies and success rates for each task are shown in Table 10.

| Task | Human | | NILS | | NILS Noisy | |
|---|---|---|---|---|---|---|
| | Grounding | Success | Grounding | Success | Grounding | Success |
| banana in sink | 1.0 | 0.67 | 1.0 | 1.0 | 1.0 | 1.0 |
| pot right | 1.0 | 1.0 | 1.0 | 1.0 | 1.0 | 0.67 |
| pot left | 1.0 | 0.67 | 1.0 | 0.67 | 1.0 | 0.33 |
| open microwave | 1.0 | 1.0 | 0.67 | 0.67 | 0.0 | 0.0 |
| open oven | 1.0 | 0.0 | 1.0 | 0.0 | 1.0 | 0.0 |
| open fridge | 1.0 | 0.0 | 1.0 | 0.0 | 1.0 | 0.33 |
| close microwave | 1.0 | 1.0 | 1.0 | 1.0 | 0.67 | 0.67 |
| close oven | 1.0 | 1.0 | 1.0 | 0.67 | 0.0 | 0.0 |
| close fridge | 0.67 | 0.33 | 1.0 | 0.67 | 0.33 | 0.33 |
| banana on stove | 1.0 | 0.33 | 0.33 | 0.0 | 0.0 | 0.0 |
| banana oven | 0.0 | 0.0 | 0.0 | 0.0 | 0.0 | 0.0 |
| pot in sink | 1.0 | 0.67 | 1.0 | 0.0 | 1.0 | 0.0 |

Table 10: Relative performance for task groundings and successful task completions. We report the average performance of all tested tasks, where the first number depicts the relative number of correctly grounded tasks, and the second the relative number of successful completions.

**Real Robot Policy.** For our experiments, we use the BESO policy architecture [57]. The model consists of a transformer architecture and uses a continuous-time diffusion generative model to generate a sequence of 20 future actions. A pretrained CLIP text encoder encodes the text instructions, while images are encoded with FiLM-conditioned ResNet-18s. We train the resulting policy on our real-robot dataset for approx. 400 epochs with a batch size of 512. Our policy learns to predict a sequence of joint state positions.

**Results.** Next, we train a language-conditioned policy [57] on the kitchen dataset and compare its performance using labels provided by NILS and human annotations. The policies are evaluated on their instruction understanding and success rate in solving natural language tasks (Table F.3). Figure 12 shows that the model trained with NILS labels performs competitively against the model trained with human annotations across 12 tasks, further demonstrating the **quality** of the labels generated by NILS.

# G    Limitations

**Perception.** NILS demonstrates the ability of off-the-shelf specialist models to annotate challenging long-horizon data. The major limitations of our framework are induced by these off-the-shelf models. Commonly used robotic environments and their contained objects are still very challenging for state-of-the-art models as most of them have not been trained on robot domain data. For instance, common evaluation environments in robotics are toy kitchens. Open-vocabulary detectors often struggle with grounding in such environments. For instance, NILS frequently detects the banana as a sponge in our toy kitchen setup. While there are models specifically applicable to the robotic domain, such as Spatial-VLM [20], RoboVQA [17] or PGBlip [58], these models are either not open-source or too specific for broader grounding applications. Lastly, the initial object retrieval might fail for abstract environments. However, this can be assessed through minimal human intervention. NILS allows users to specify objects they expect to appear in the scene, which would improve performance in these scenarios. Given the modularity of NILS, better models also result in a higher grounding accuracy.

**Runtime.** Using multiple different models to generate scene representations introduces substantial computational cost. The inference time of our framework is significantly higher compared to tested baselines. NILS requires 7 minutes to label one long-horizon trajectory consisting of 50 tasks on a single 3090 RTX GPU. The framework is designed to be applied offline to prerecorded play data, thus the higher runtime should be manageable.

**Accuracy.** While our framework shows good performance for different scenes and objects, it sometimes produces wrong labels. Although the heuristics used to detect keystates can filter noise effectively, they are sometimes correlated. In such cases, noise triggers all keystate heuristics, resulting in a high confidence keystate. Furthermore, NILS is designed for long-horizon demonstrations. To generate robust scene annotations, NILS relies on temporal consistency and diversity, which is not always given in short horizon demonstrations.

**Object-Centric Trade-off** NILS has high objectness assumptions for accurate labeling. This makes it challenging to label tasks with granular objects such as rice corns. In future work, we want to explore how to better integrate the VLM in such scenarios. Additionally, due to the state of object detectors, we can not distinguish between objects that are very similar in appearance, such as similar blocks.

# H    Extended Related Work

## H.1    Learning from Play

Robots usually learn from short-term demonstrations of a single task, but this approach has drawbacks such as lack of diversity, inefficient data collection [17], and poor generalization. As an alternative, Learning from Play [59] proposes collecting unconstrained operator interactions in the environment, resulting in more diverse data and better generalization. Extracting goal states from these long demonstrations is necessary for goal-conditioned imitation learning. While sampling random windows or using robot proprietary information can provide image goals [59, 60, 11], extending to language goals is challenging as they must align with observations. Most methods use joint goal spaces to embed multiple modalities [2, 61, 49, 62, 57, 63], but still require some language annotations. Our approach densely predicts actions performed in long-horizon trajectories to

address this issue. Long-horizon datasets with annotated languages, like CALVIN [2] or TACO Play [40, 64], are scarce due to these challenges.

# I  Keystate Visualizations

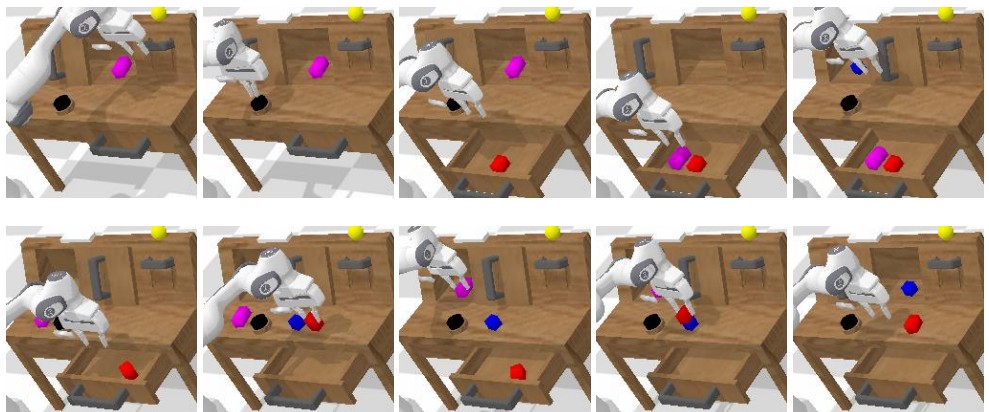

Figure 13: Visualization of keystates extracted by NILS for the CALVIN environment. Keystates are extracted only via RGB Images, without incorporating gripper-close signals. In the second sequence, some intermediate keystates are not detected. Nevertheless, the predicted keystates are precise.

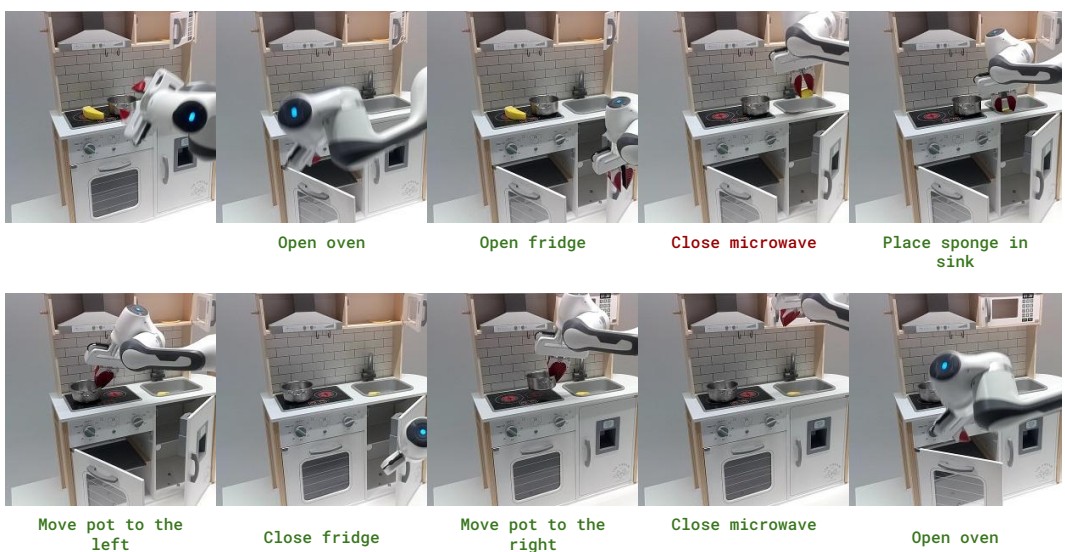

Figure 14: Example trajectory labeled with NILS. NILS successfully detected the keystates in the long-horizon trajectory and correctly grounds the performed action for tasks.

## J  Foundation Model Prompts

```
<Frame 1>...<Frame 8>
Given the image sequence, what task did the robot perform? Output the task the robot
performed as a short instruction with three paraphrases, delimited by comma.
The task should include the interacted object and the object's resulting relation to
other objects.
Example tasks: "Place the pot to the left of the fork", "Move the dishrag to the bottom
of the table next to the blue towel", "Pick up the cucumber and place it to the right
of the bowl", "Turn on stove", "Open the microwave", "Put the knife inside the sink".
Output valid json in the following format:
{ "tasks": "the predicted tasks.  The tasks should always include the final object
positions if possible."}
```

Figure 15: VLM Baseline Prompt

```
<Image >
Create a json list where each entry is an object in the image. The entry should have
the keys "name" with a unique, concise, up to six-word description of the object and
"color", containing the object color. Always output the surface the objects are placed
on as the first entry. If the surface is unknown, output "None".
An example:
'''
[{"name": pot with handle,
"color": "silver"}]
'''
```

Figure 16: Stage 1 Object Retrieval Prompt.

```
You will be provided with a list of objects observed by a robot. Based on the
objects, give possible instructions to the robot. Infer the type of environment
from the provided objects. Follow these guidelines:

- Keep the instructions simple.  Focus on tasks that only require a single
step.
- Include tasks like placing an object inside another object or moving the object.
Only for movable objects.
- Dont assume the presence of any objects not listed.
Output at least 20 possible instructions delimited by comma.

Here are a few examples:  "Place the tin can to the left of the pot.",
"Move the dishrag to the bottom of the table next to the towel","Put the pot to
the right of the fruit","Turn on stove", "Open the microwave"

The following objects are in the environment: [OBJECT_LIST]
```

Figure 17: Task generation prompt

```
You will be provided with observations of a robot interaction with an environment,
delimited by triple quotes.
Determine the task the robot could have solved. The robot can only solve one task.
If the observations indicate that the robot interacted with multiple objects,
focus on the most frequent and precise observations.

Follow these guidelines:
Step 1: Answer what objects appear in the observation. List all objects. Then,
determine the object for which the observations align the best.
Step 2: Determine the object movement and the resulting object relations. Think
about where the object and its relational objects are located in the scene on
a global scale. Think step by step and list the locations and relations of all
objects. Explain the object movements.
Step 3: Determine what tasks result in the object relations from Step 2.
Step 4: Output tasks that that accomplish the observations as short instructions.
Focus on simple, single-step tasks that only require interaction with the determined
object from Step 1. Focus on tasks that include changing the object relation and
moving the object.
Example tasks: "Place the pot to the left of the fruit"; Slide the dishrag to the
bottom of the table next to the towel"; Pick up the spoon and place it at the
bottom left of the table;"Put the pot to the right of the fruit"; "Move the pot
forward and to the left"; "Turn on stove"; "Open the microwave"; "Relocate the
knife inside the sink". Follow the steps above. Explain your reasoning. Output the
reasoning delimited by ***.

After, produce your output as JSON. The format should be:
'''{ "tasks":  "The  determined  tasks,  delimited  by  semicolons.   Output 4
different,diverse task instructions. The instructions should cover all observations
and each include different observations. Example: Place the pot to the left of the
fruit; Move the pot backward and to the right; Relocate the pot at the left of the
table to the center of the table; Lift up the pot and place it next to the spoon;",
"confidence": "A confidence score for each task between 0 and 10, delimited by
commas. Be pessimistic." }'''

Observations: '''[OBSERVATIONS]'''
```

Figure 18: Main action retrieval prompt

# K  Play Kitchen Dataset Description

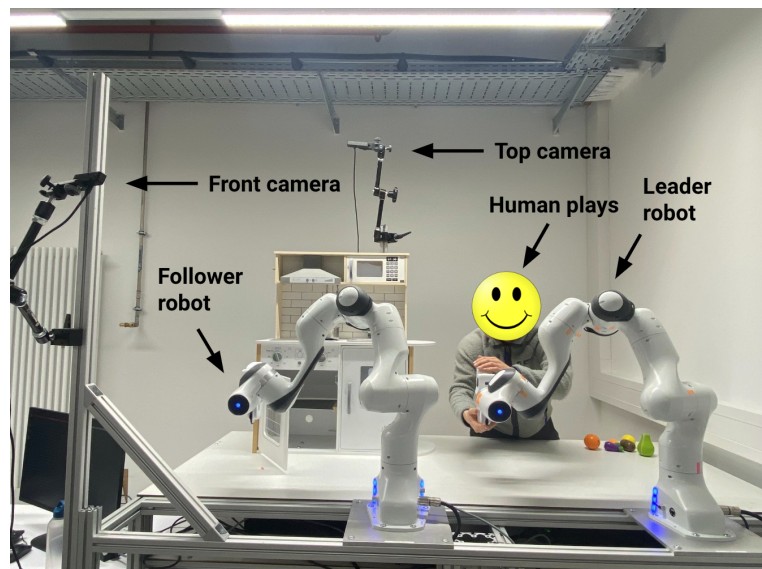

Figure 19: Overview of the teleoperation setup in the real kitchen environment. The human operates on the leader robot. The follower robot imitates the actions of the leader. The top and front cameras record the play trajectory at 30Hz.

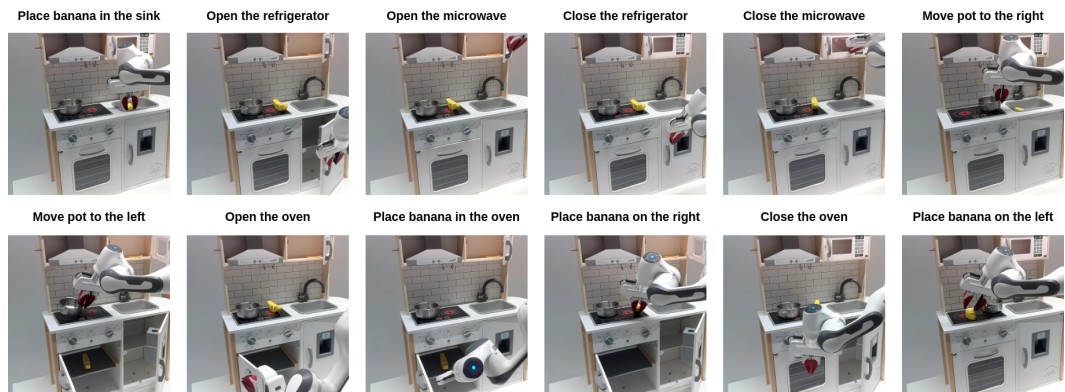

Figure 20: Overview of the 12 tasks recorded during play from the preprocessed front camera perspective.

# L  Comparison to Gemini Vision Pro on BridgeV2

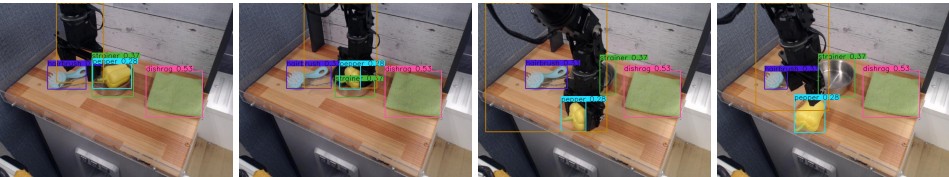

**Instructions generated by NILS:** "Move the pepper from inside the strainer to in front of the strainer", "Take the pepper out of the strainer and place it forward", "Relocate the pepper to a position in front of the strainer"

**Instructions generated by Gemini Vision Pro**: "Move the yellow bell pepper to the left", "Place the yellow bell pepper in the pot", "Move the pot to the right."

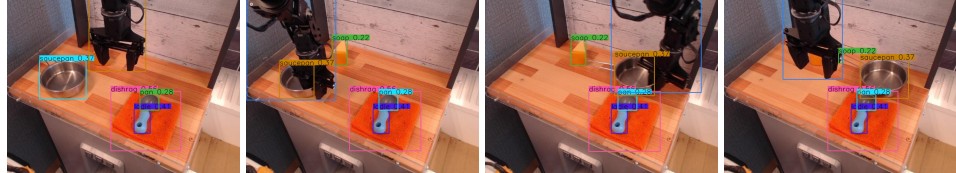

**Instructions generated by NILS:** "Place the saucepan on top of the dishrag", "Move the saucepan to the right of the soap", "Position the saucepan behind the ladle."

**Instructions generated by Gemini Vision Pro**: "Move the cheese to the right", "Move the bowl to the right", "Move the spoon to the right", "Move the dishrag to the right."

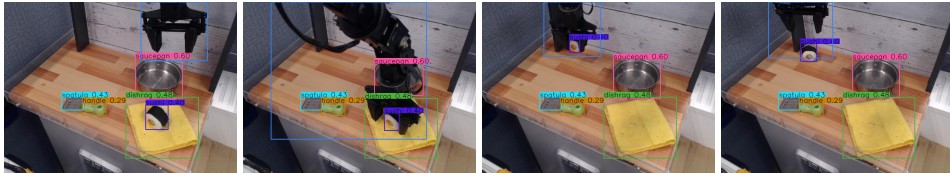

**Instructions generated by NILS:** "Move the sushi from on top of the dishrag to a new location away from the saucepan", "Relocate the sushi to clear the area on top of the dishrag", "Shift the sushi to organize the workspace, ensuring it is no longer next to the saucepan."

**Instructions generated by Gemini Vision Pro**: "The robot moved the green spatula from the left of the cutting board to the right of the cutting board", "The robot moved the yellow cloth from the right of the cutting board to the left of the cutting board",

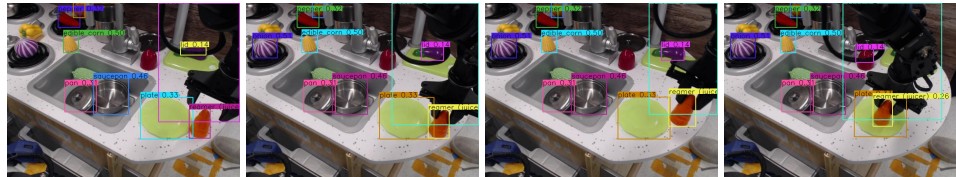

**Instructions generated by NILS:** "Put the reamer (juicer) inside the plate", "Move the reamer (juicer) from next to the plate to inside it", "Place the reamer into the plate for storage or preparation"

**Instructions generated by Gemini Vision Pro**: "The robot picked up a carrot that was resting on a green plate and placed it in the sink.", "The robot moved a carrot from a green plate to the sink."

Figure 21: Comparison of generated labels for samples form the bridge V2 dataset. We show the labels generated by NILS and Gemini Pro for several tasks.

