# OpenReview forum: "Scaling Robot Policy Learning via Zero-Shot Labeling with Foundation Models"
_robot-learning.org/CoRL/2024/Conference — CoRL 2024_

### Official Review · Reviewer_B6Tb · 2024-07-14
**novel, however overly complicated with limited evaluation datasets**

**Originality:** 3
**Technical Quality:** 3
**Clarity Of Presentation:** 3
**Potential Impact:** 3
**Recommendation:** 3
**Confidence:** 4

**Review:**

The paper presents an interesting and potentially impactful approach to automatically labeling robot demonstrations for language-conditioned policy learning. The use of multiple pre-trained foundation models to enable zero-shot labeling is novel and addresses a key challenge in scaling up robot learning. The preliminary results on BridgeV2 and the custom kitchen dataset are promising, showing NILS can generate reasonable labels and train policies competitive with human-annotated data.

However, the current framework seems overly complex, utilizing multiple vision models, LLMs, and VLMs. It's unclear if this level of complexity is necessary or if a simpler approach could achieve similar results. The paper would benefit from ablation studies and comparisons to simpler baselines to justify each component.

The evaluation is somewhat limited, using relatively small subsets of data with low task and scene diversity. A more comprehensive evaluation on larger, more diverse datasets would strengthen the claims. Direct evaluation of label quality and correctness would also be valuable.

Strengths:
1. Novel approach to zero-shot labeling of robot demonstrations using an ensemble of pre-trained models. Addresses an important challenge in scaling up language-conditioned robot learning.

2. Preliminary results show promise for generating high-quality labels without human annotation Enables training language-conditioned policies on unlabeled data.

3. Modular design allows incorporating improved foundation models.

Weaknesses:
1. The framework seems overly complex, using multiple vision models, LLMs, and VLMs - it's unclear if this level of complexity is necessary.

2. Limited evaluation on relatively small datasets with low task/scene diversity.

3. Lack of comparison to simpler baselines or ablations justifying each component

**Quality Of The Limitations Section:**

1

**Questions For Rebuttal:**

1. Can the framework be simplified while maintaining performance? What is the justification for each component?

2. How does NILS perform on larger, more diverse datasets?

3. How does NILS compare to simpler baselines or individual foundation models for this task?

4. Can you provide results on label quality?

**Robotics Focus:**

4

**Summary Of Paper:**

This paper introduces NILS, a framework for automatically labeling long-horizon robot demonstrations without human annotations. NILS uses an ensemble of pre-trained vision-language models to detect objects, segment tasks, and generate natural language labels for uncurated robot data. The key novelty is leveraging multiple frozen foundation models to enable zero-shot labeling of diverse robot behaviors. The authors evaluate NILS on a subset of BridgeV2 and a custom kitchen dataset, showing it can generate high-quality labels and train language-conditioned policies competitive with human-annotated data.

**Summary Of Recommendation:**

I recommend a weak accept for this paper. The proposed NILS framework presents a novel and potentially impactful approach to zero-shot labeling of robot demonstrations using pre-trained foundation models. While the preliminary results are promising, the framework's complexity, limited evaluation, and lack of thorough ablations are significant weaknesses. However, the core idea is valuable and addresses an important challenge in scaling language-conditioned robot learning. With revisions to address the identified weaknesses, particularly simplifying the framework and expanding the evaluation, this work could make a strong contribution to the field.

---

### Official Review · Reviewer_PVKm · 2024-07-18

**Originality:** 3
**Technical Quality:** 4
**Clarity Of Presentation:** 3
**Potential Impact:** 4
**Recommendation:** 3
**Confidence:** 4

**Review:**

**Strengths**:
- The motivation behind the paper is strong and its standing with respect to the literature seems fitting.
- They are intelligently able to leverage the tractability and ease of use of templated descriptions while still ultimately obtaining diverse descriptions thanks to the use of LLMs
- While systems papers might be less relevant to the research community, they do a plethora of experiments which are tailored to their work and therefore provide useful insights.

**Weaknesses**:

Overall, I believe that even though the clarity of the paper is not bad, it is also not a strength and it could be improved.
- The method section is in my opinion excessively convoluted. The method itself is fairly simple to understand, however you have rightfully put together quite a few heuristics and systems to identify or filter out different objects and object properties. This is more than fine, but you end up with a section that is way too cluttered and not so detailed. You try to discuss every single design choice you make, but at the expense of only spending one or maximum two sentences on each, which turns out to be chaotic. I discuss in the rebuttal section my suggestions.

- Throughout the paper you mention terms like: "long-horizon demonstrations", "short-horizon demonstrations", "task" and "sub task" without ever defining what these are for you. For instance on line 191 "a self-collected one hour long-horizon, uncurated play dataset ... containing 439 short-horizon demonstrations of 12 tasks". Is a long horizon demonstrations one without cuts, or even a chained sequence of shorter videos? Is a short horizon demo, one for a single task? Is a "task" a single stage or is that a sub task, for example is pick and place a task while the individual picking and the individual placing are sub tasks? If that is so, is a short horizon demo a demo only for a sub task, while a demo of a task (i.e. multiple sub tasks) is a long-horizon demo? All these questions impact the overall clarity of the paper and should be addressed.

- In the experiments section:
   - I think that the discussion of the various experiments is often limited to simply stating what can be very quickly understood from the tables or figures. A more tailored discussion of the motivation behind certain design choices as well as what was found to be hard or easy for your method to do would in my opinion improve the overall contribution.
    - Some parts of the experiments are unclear to me. For instance on line 219 you say that "frames (are) uniformly sampled from a short horizon demonstration using the keystates from our method" and then in Table 1 you say you used GT keystates. Which one is it, are they (1) uniformly sampled or (2) do you use the keystates predicted by NILS or (3) do you use GT keystates from human annotations?
    - Another unclear part is Section 3.2 where you do not explain how you calculate the accuracy metric for the grounding performance.
    - In section 3.4 you do not mention what "Correct Grounding" is. I know it is explained in the Appendix, but the figure is in the main body so you should explain it there too.

**Originality**:
- The individual components of the system have been proposed before. However, the way these components have been put together here is clever and previously unseen.

**Significance**:
- Work in the direction of auto labelling has the potential to have a great impact on the community. With energy costs going down, compute power going up, and foundation models constantly improving, methods like these also have the ability to scale.

**Quality Of The Limitations Section:**

3

**Questions For Rebuttal:**

I want to be clear that I believe all the necessary material for the paper is definitely there. I only think it would be wise to address the overall clarity of the paper.

**Weakly Suggest**:
- In the method section I think it would be good to filter out some of the things you discuss and move them to the appendix, where you can discuss them in as much detail as possible. In the actual method section I would discuss the system on a higher level. The sections that would benefit the most are "Stage 1: Identifying Objects in the Scene", "Object Annotations and Segmentations" and "Object Relations and Object Movement". It is perfectly fine in my opinion if the method section becomes shorter, it would just free up some space for more discussion of your findings or moving an additional experiment up from the appendix.
- Try discussing more insightful ideas or conclusions in the experiments section.

**Strongly suggest**:

Address the few things mentioned in the Review above.

**Minor Corrections**:
- Figures 3 and 4 should be flipped
- Figure 12 is before Figure 11
- Appendix K is empty because it starts after the relevant figure.

**Robotics Focus:**

4

**Summary Of Paper:**

With this paper the authors are introducing NILS, a system that can be used to label long-horizon robot videos with diverse language annotations. They achieve this by sequentially using an ensable of VLMs that processes the videos in an object centric way. At a high-level this process is divided into 3 stages. The first stage consists in detecting the objects in a scene in a temporally consistent way, filtering redundancies and classifying the objects' properties. The second stage tracks each object's movement and relative relations among other things and defines these as templated language descriptions. Finally, stage 3 identifies the keyframes in the video and provides the templated information for the relevant frames to an LLM which combines and merges these to create a set of diverse language descriptions of variable granularity. Additionally, they evaluate this system on a couple of annotated datasets as well as testing the quality of the labels by training a real world policy with them.

**Summary Of Recommendation:**

This work can potentially help the community to scale data labelling without the need for humans. The information needed to understand the underlying system is all there. They present enough experiments that can provide insights on the performance and usefulness of the method. However, in order to have a strong enough paper I advise working on the clarity of the manuscript as well as focusing more on the experiments.

---

### Official Review · Reviewer_p3J4 · 2024-07-21
**Review for NILS**

**Originality:** 3
**Technical Quality:** 2
**Clarity Of Presentation:** 4
**Potential Impact:** 3
**Recommendation:** 2
**Confidence:** 3

**Review:**

Strengths:
- This paper presents an interesting approach to acquiring useful labels using pre-trained foundation models. The resulting labels are more fine-grained with the approach than naively using VLMs. The approach is also autonomous and does not require any additional human effort, which is a large bottleneck for language-conditioned datasets.
- The paper is well written and clearly organized. It properly addresses limitations of the approach.

Weaknesses:
- I think that one of the most important evaluations for this paper to be convincing is showing that the downstream policy performance trained with this labeled data performs well. The paper provides an initial evaluation of this in Section 3.4 but only compares to human annotations. How do NILS and NILS noisy compare to Gemini and GPT-4o in this evaluation setting?
- The labeling process seems quite time-consuming, taking up to 7 minutes per long-horizon trajectory.
- Foundation models can be inaccurate in complex scenes, and including noisy examples seems to hurt downstream performance. Might need some better label filtering to work for more difficult tasks.

**Quality Of The Limitations Section:**

3

**Questions For Rebuttal:**

See weaknesses above. Also:
- For NILS vs NILS Noisy, how do you determine what is an ambiguous label?

**Robotics Focus:**

4

**Summary Of Paper:**

This paper aims to address the challenge of labeling natural language instructions in robot datasets. To do so, it proposes NILS (Natural language Instruction Labeling for Scalability), which uses pre-trained VLMs to identify objects, detect changes, and segment tasks. The authors evaluate on BridgeV2 and a kitchen play dataset and finds that the annotations lead to comparable policy performance compared with human annotations.

**Summary Of Recommendation:**

The current state of the paper requires some additional experiments and analyses to be convincing. If my concerns are addressed, I will be happy to bump up my recommendation.

---

### Author Rebuttal · Authors · 2024-08-12

We thank all the reviewers and AC for their thoughtful comments, and we are glad to see that our work was received positively!

The comments and suggestions have helped us significantly to further improve the quality and clarity of the paper.
We have responded to each reviewer individually below.

To summarize, we added additional real world policy evaluation on the Bridge setup [1], as well as further experiments using the Simpler Benchmark [2] in Section 3.4, which is a sim benchmark tailored for evaluating policies trained on real world data, to strengthen our evaluation.
Our findings remain consistent: our experiments show that policies trained with NILS labels match the performance of those trained on crowd-sourced labels in both real robot experiments and in simulation.

We additionally have revised the paper to further clarify and simplify the details of the method. Please see the attached PDF for the revisions.

We employed color coding, so each reviewer can easily identify the changes we made to the manuscript according to their recommendations:

Reviewer p3J4 - blue

Reviewer PVKm - orange

Reviewer B6Tb -purple

Please note that Reviewer p3J4 and  B6Tb showed similar concerns regarding updated experiments and ablations, which is why there is a high overlap between colors.



[1] Walke, Homer Rich, et al. "Bridgedata v2: A dataset for robot learning at scale." Conference on Robot Learning. PMLR, 2023.

[2] Li, Xuanlin, et al. "Evaluating Real-World Robot Manipulation Policies in Simulation." arXiv preprint arXiv:2405.05941 (2024).

---

### Decision · Program_Chairs · 2024-09-04

**Decision:**

Accept

**Comment:**

This work proposes a language annotation strategy for unlabelled video demonstrations using foundation models. They propose a factored approach to generate annotations taking into consideration objects in the scene, their relationships etc. Authors are encouraged to improve the technical exposition in relation to the effectiveness of the annotation strategy in facilitating downstream policy learning, scalability of the method and include comparison with other models.